# Divide and Conquer: Isolating Normal-Abnormal Attributes in Knowledge Graph-Enhanced Radiology Report Generation

## ABSTRACT

Radiology report generation aims to automatically generate clinical descriptions for radiology images, reducing the workload of radiologists. Compared to general image captioning tasks, the subtle differences in medical images and the specialized, complex nature of medical terminology limit the performance of data-driven radiology report generation. Previous research has attempted to leverage prior knowledge, such as organ-disease graphs, to enhance models' abilities to identify specific diseases and generate corresponding medical terminology. However, these methods cover only a limited number of disease types, focusing solely on disease terms mentioned in reports but ignoring their normal or abnormal attributes, which are critical to generating accurate reports. To address this issue, we propose a **D**ivide-and-**C**onquer approach, named DCG, which separately constructs disease-free and disease-specific nodes within the knowledge graphs. Specifically, we extracted more comprehensive organ-disease entities from reports than previous methods and constructed disease-free and disease-specific nodes by rigorously distinguishing between normal conditions and specific diseases. This enables our model to consciously focus on abnormal information and mitigate the impact of excessively common diseases on report generation. Subsequently, the constructed graph is utilized to enhance the correlation between visual representations and disease terminology, thereby guiding the decoder in report generation. Extensive experiments conducted on benchmark datasets IU-Xray and MIMIC-CXR demonstrate the superiority of our proposed method. Code is available at the anonymous repository[1].

## CCS CONCEPTS

• **Computing methodologies** → *Natural language generation*; *Image representations.*

## KEYWORDS

Radiology Report Generation, Image Captioning, Medical Image Analysis, Vision and Language

## 1 INTRODUCTION

Radiology images, such as chest X-rays, are crucial in routine diagnosis and treatment. When analyzing a radiology image, radiologists need to assess both normal and abnormal types in each region,

---

[1]https://anonymous.4open.science/r/DCG_Enhanced_distilGPT2-37D2

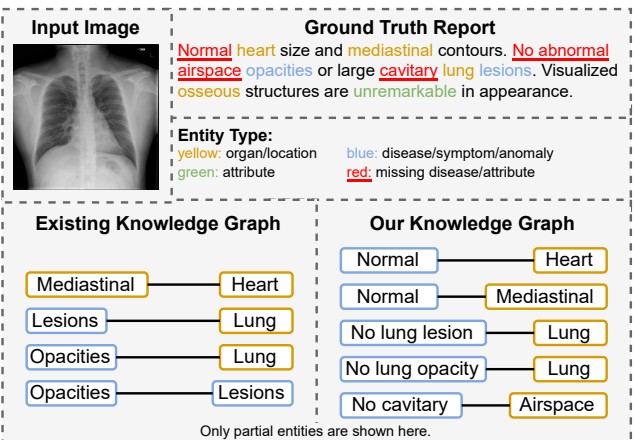

**Figure 1: Example of a Chest X-ray and report from IU-Xray, showcasing the organ-disease knowledge graph (KG) used by existing methods, alongside our newly reconstructed KG. Yellow, blue, and green in the report denote different entity types. Red underlines highlight diseases or attributes overlooked by existing methods. The KG triplets generated by the representative method DCL [1] can be downloaded from[2].**

drawing upon their professional knowledge and clinical experience to document these findings, a process that is often time-consuming. The advancement of automatic radiology report generation systems [2] has the potential to significantly reduce radiologists' workload, garnering keen interest from both the medical and computer science communities. Thanks to the substantial advancements in artificial intelligence, particularly in deep learning methodologies, various data-driven neural networks designed for radiology reports have been proposed [3, 4]. Among them, encoder-decoder architectures based on attention mechanisms [5] have been widely adopted, achieving promising performance. However, these data-driven methods face challenges due to the following data biases: 1) Descriptions of common diseases dominate, while descriptions of uncommon diseases are rare. 2) Reports annotated by experts tend to emphasize abnormal descriptions and often omit or briefly describe normal conditions. 3) The radiology images have similar appearances, and the key features determining whether they have abnormalities are often subtle.

These challenges require the model to accurately discern subtle differences in images, counteract the effects of data bias, and ensure comprehensive report generation. To this end, recent research has focused on integrating prior medical knowledge into the report generation task, enhancing the model's ability to generate medical terminology and distinguish specific diseases. For example, MKG

---

[2]https://github.com/mlii0117/DCL

[6] extracted seven organs/tissues and twenty disease findings from radiology reports to serve as nodes in a graph, constructing a universal graph by connecting specific organs and diseases to enhance the relationship between different areas and abnormal findings. The graph built by MKG [6] has been widely adopted in subsequent research. PPKED [7] and DCL [1], building upon MKG, expanded the scope of knowledge further by leveraging prior knowledge from previously retrieved radiology reports in the training corpus. Although some success has been achieved in detecting potential specific diseases, these methods still face the following limitations:

- **Limited disease coverage.** Figure 1 shows a radiology report and an organ-disease graph predefined by existing methods. The entity "cavitary", often associated with lung infections like tuberculosis or specific types of pneumonia, is crucial for diagnosis but overlooked.
- **Overlook distinguishing between normal and abnormal attributes.** The existing graph in Figure 1 mainly focuses on the presence of disease entities such as "Lesions" in a report while neglecting critical medical attributes indicated by terms like "No" or "Normal". Despite some n-gram overlap of the generated reports with the ground truth, these approaches still falls short in identifying key medical attributes (normal or abnormal), limiting their clinical applicability.

To address the aforementioned issues, we have innovated in organ-disease graph construction and proposed a novel **D**ivide-and-**C**onquer approach called DCG, which not only focuses on specific organ-disease relationships but also emphasizes the normal and abnormal attributes of diseases, ensuring a high consistency with actual report descriptions. Specifically, we first utilize the powerful biomedically pre-trained foundation model, BioMedCLIP [8], as a retriever to retrieve the top-$K$ similar radiology images from the training set for each input image, and obtain their corresponding reports. Then, we extract fine-grained disease entities from these reports, further subdividing common diseases such as "Lesion" into "Lobe lesion" or "Bone lesion". Following a divide-and-conquer strategy, we categorize them strictly as *disease-free* or *disease-specific*, based on their normal or abnormal attributes as well as specific locations, such as "Lobe lesion" or "No bone lesion". Subsequently, these text entities are used to construct nodes in the graph, with a graph convolutional network being employed to model the unique relationships between nodes within each retrieved report. The resulting node embeddings are then utilized to enhance the fine-grained patch embeddings extracted by the image encoder, thereby improving the association with specific disease texts and ultimately guiding the generation of radiology reports. Our contributions can be summarized as follows:

- We proposed a **D**ivide-and-**C**onquer strategy, named DCG, that constructs a more comprehensive organ-disease graph encompassing a broader range of diseases than previous methods, serving as prior knowledge to enhance radiology report generation.
- The proposed **D**ivide-and-**C**onquer approach categorizes entities as *disease-free* or *disease-specific* based on their attributes, effectively reducing the impact of common diseases and distinguishing between normal and abnormal conditions.

- Extensive experiments conducted on benchmark datasets IU-Xray [9] and MIMIC-CXR [10] demonstrate the superiority of our proposed method.

## 2 RELATED WORK

### 2.1 Image Captioning

Image captioning aims to generate coherent and accurate natural language description for an image. It relies on identifying entities in the image and appropriately describing the relationships between them. Bridging the semantic gap between different modalities remains a challenging task, but the impressive achievements of the transformer in natural language processing and multimodal domains have notably propelled the progress of image caption methods [11–15]. Representative methods such as OSCAR [16] utilize object tags detected in images as anchor points for aligning images and language, thereby simplifying the learning process of semantic alignment between image and text. UpDown [17] employs a bottom-up mechanism to extract interested image regions and image features, and a top-down mechanism is used to learn to adjust feature weights. CAAG [18] first generates global context using the primary captioning model and then selectively generates target words at each time step based on the global context and hidden states. Particularly, considering the complex spatial and semantic interactions between objects in images and entities in sentences, recent efforts have utilized scene graphs [19] to model the objects in images for better semantic alignment from vision to language, ultimately achieving accurate image captioning. For instance, TFSGC [20] projects scene graphs as token sequences and introduces binary masks to index connected nodes in the graph, with learnable type embeddings aiding the model in distinguishing types of edges, thereby generating more precise captions. K-Replay [21] selects a small number of images containing knowledge as enhancement examples and employs a sentence-level knowledge coverage loss to prevent model collapse during fine-tuning, successfully integrating the generalization capabilities of vision-language pre-trained models into image captioning. Despite challenges in data collection and the need for professional annotations in radiology report generation and the broader biomedical multi-modal domain, there currently exists no large-scale scene graph dataset akin to Visual Genome [19] for detailed modeling of object entities in images. However, we are diligently working to bridge this data gap using structured knowledge like organ-disease graphs, coupled with innovative and adaptive methods, to enhance the accuracy and clinical applicability of radiology report generation.

### 2.2 Radiology Report Generation

Radiology report generation (RRG) task aims to generate clinical descriptions for radiological images and is often considered an extension of image captioning in the medical domain. Inspired by image captioning, most RRG models rely on an encoder-decoder architecture for report generation [1, 2, 6, 7, 22–27]. However, compared to general domain image captioning, RRG faces two primary challenges: medical reports are much longer than generic image captions, and the high similarity in the appearance of radiology images makes detecting subtle abnormal information more difficult than identifying objects in general images. Many approaches have

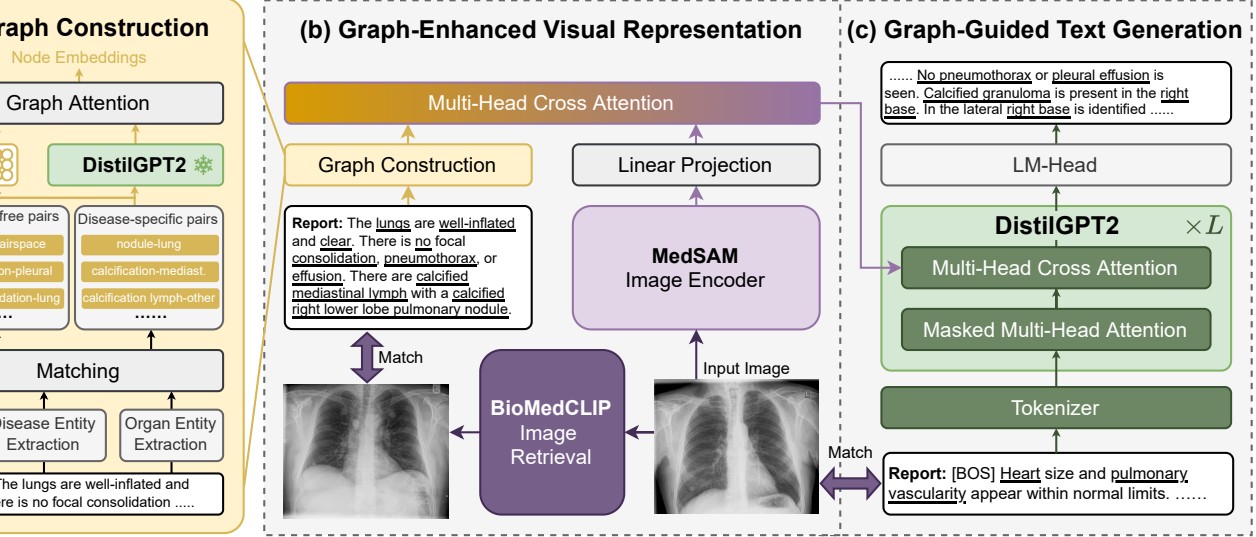

**Figure 2: Overview of our Divide-and-Conquer approach, which consists of three modules: (a) Divide-and-Conquer Graph Construction, (b) Graph-Enhanced Visual Representation, and (c) Graph-Guided Text Generation. Each input image is paired with a corresponding report acquired through an offline image retrieval process. The retrieved report is then input into module (a) to construct a unique graph. This graph is utilized in module (b) to enhance visual representations. Module (c), in an autoregressive manner, generates each subword of the radiology report based on the previously produced subwords and the enhanced visual representations. [BOS] symbolizes the beginning-of-sentence special token.**

been proposed to address these challenges. For example, Liu et al. [22] built upon the standard transformer framework and proposed a progressive generation architecture, which first predicts the preliminary topics of reports and then generates sentences corresponding to these topics. R2Gen [2] added a relationship memory network to automatically record the generation process and used the recorded information to guide report generation. CMN [23] employed a memory matrix to achieve cross-modal alignment and interaction.

Due to the inherent characteristics of radiology images, where a strong intrinsic connection between diseases and organs exists, recent approaches using graphs indicate the intrinsic correlations between diseases and organs to assist in report generation. For example, KERP [24] extracted anomalies from radiology images to construct an anomaly graph and transformed high-level semantics across various domains of graph-structured data through a graph transformer. MKG [6] constructed a graph consisting of 7 organs/tissues and 20 findings from numerous findings in reports, with edges describing the relationships between findings and organs/tissues. Following [6], PPKED [7] combined prior knowledge with the pre-constructed graph to refine knowledge. DCL [1], building on a pre-constructed organ-disease graph, retrieves entities during the generation process as supplements to the graph, thereby dynamically expanding the graph's knowledge scope. Compared to existing methods with only 7 organs/tissues and 20 findings [1, 6, 7], we further subdivide diseases according to the fine-grained tissues where they occur, such as subdividing "Lesion" into "Lobe

lesion" or "Bone lesion". Following our proposed Divide-and-Conquer strategy, we strictly categorize the normal and abnormal attributes of disease descriptions, such as "Lobe lesion" or "No lobe lesion", to construct the organ-disease graph. Our constructed graph expands the breadth of knowledge coverage and provides the decoder with information more consistent with the semantics of expert-written reports.

## 3 METHODOLOGY

In this section, we will introduce the detailed implementation of our proposed **D**ivide-and-**C**onquer **G**raph-enhanced radiology report generation. An overview of the structure of DCG is illustrated in Figure 2, which includes three modules: (a) Divide-and-Conquer Graph Construction, (b) Graph-Enhanced Visual Representation, and (c) Graph-Guided Text Generation. We first present the optimization objective for radiology report generation and then sequentially introduce the three proposed modules.

### 3.1 Overview

The radiology report generation task aims to produce a textual sequence $\mathcal{Y} = \{y_1, \ldots, y_T\}$, which describes the input radiology image $\mathcal{I}$. Here, $y_t$ represents a word token in the report, and $T$ is the length of the report. The entire process of generating the textual sequence can be expressed as:

$$P(\mathcal{Y}|\mathcal{I}) = \prod_{t=1}^{T} P(y_t|y_{<t}, \mathcal{I}). \quad (1)$$

Here $t$ indexes each token in the sequence, while $y_{<t}$ represents all the tokens that precede the $t$-th token in the generated

report. Typically, cross-entropy loss is used to optimize the model parameters $\theta$:

$$\mathcal{L}_{CE}(\theta) = -\sum_{t=1}^{T} \log P_\theta(y_t^* | y_{<t}^*, \mathcal{I}), \quad (2)$$

where $\mathcal{Y}^* = \{y_1^*, \ldots, y_T^*\}$ represents the ground truth report. In this paper, we follow the standard image captioning structure of an image encoder and an auto-regressive text decoder, while also utilizing the graph to enhance visual representation, further guiding the accurate generation of radiology reports.

## 3.2 Divide-and-Conquer Graph Construction

To utilize prior knowledge to enhance image representation and guide the generation of radiology reports, we first need to construct a organ-disease graph. It is worth emphasizing that the organ-disease graph [6, 7] used in radiology report generation differs from common knowledge graphs, exclusively describing the relationships between organ and disease entities. Unlike the approach in [28], which utilizes an image classifier pre-trained on the [29] or [30] dataset to predict symptom labels for graph construction, we employ a retrieval-augmented approach that directly retrieves the image $\mathcal{I}'$, which is similar to the input image $\mathcal{I}$, and use its corresponding report to construct the graph. Specifically, we first utilize the image encoder (ViT) of BioMedCLIP [8] to initialize the set of image indices $\mathcal{D} = \{f_{\mathcal{I}_i}\}_{i=1}^{L}$, where each $f_{\mathcal{I}_i}$ is the [CLS] embedding extracted for the $i$-th image by the image encoder and $L$ denote the total number of images. Subsequently, the Top-$K$ items retrieved from $\mathcal{D}$ based on the highest cosine similarity to a given image query $f_{\mathcal{I}}$ are represented as:

$$\tilde{\mathcal{D}} = \text{Top-K}(\mathcal{D} \,|\, f_{\mathcal{I}}), \quad (3)$$

where $\tilde{\mathcal{D}} \subseteq \mathcal{D}$, and the corresponding reports $\mathcal{Y}'$ are obtained by concatenation [; ], resulting in:

$$\mathcal{Y}' = [\mathcal{Y}_1; \ldots; \mathcal{Y}_{|\tilde{\mathcal{D}}|}]. \quad (4)$$

Afterward, $\mathcal{Y}'$ is used to extract organ-disease entities. The predefined organ and disease entities, along with their detailed relationships, are illustrated in Figure 3. Our predefined organs are consistent with those in R2Gen [2], but in addition to symptoms specific to each organ (such as "Pneumonia" in "Lung" and "Fracture" in "Bone"), our predefined symptoms are strictly distinguished by the organ in which they occur, for example ("Rib lesion" in "Bone", "Lobe lesion" in "Lung"), to avoid the confusion and potential inaccuracies that may arise from report generation. To further differentiate between normal and abnormal conditions in radiology images and achieve precise matching with ground truth reports, we employ a divide-and-conquer strategy to construct two types of disease-organ pairs using report text, organs, and detailed symptoms: disease-free and disease-specific pairs, as illustrated in the lower part of Figure 3. Throughout the entire dataset, disease-specific and disease-free entities are designated as $N$ nodes $\mathcal{V} = \{v_1, v_2, \ldots, v_N\}$ in the Divide-and-Conquer Graph $\mathcal{G} = (\mathcal{E}, \mathcal{V})$. Node relationships include "exists" and "does not exist", and entities such as "Effusion" for disease-specific and "No Effusion" for disease-free cases are encoded into node embeddings $\mathbf{F}_v$ using pre-trained DistilGPT2 [31]. We take the average

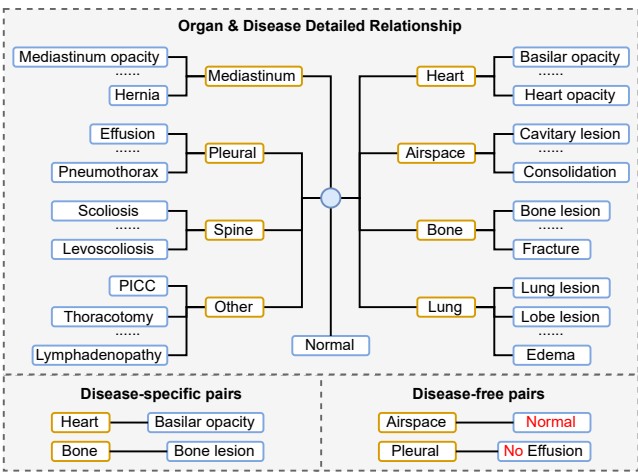

**Figure 3: Upper part: Predefined organs, detailed diseases/symptoms, and their relationships; Lower part: Examples of disease-specific pairs and disease-free pairs.**

of the last hidden states from the final layer of the transformer to obtain the set of all node embeddings $\mathbf{F}_v = \{f_{v_i} \in \mathbb{R}^d\}_{i=1}^{N}$. If an organ-disease-specific or an organ-disease-free condition is present in the retrieved report $\mathcal{Y}'$, their relationship is represented by an edge $e_j$, where $\mathcal{E} = \{e_j\}_{j=1}^{M}$, and $M$ denotes the number of edges. A $U$-layer Graph Convolutional Network (GCN) [32] with residual connections is employed to model the graph:

$$\mathbf{F}_v^{(u)} = \text{GeLU}\left((\mathbf{R} + \mathbf{I})\mathbf{F}_v^{(u-1)}\mathbf{W}^{(u)}\right), \quad (5)$$

where GeLU is the activation function, $\mathbf{R} \in \mathbb{R}^{N \times N}$ is the adjacency matrix constructed from $\mathcal{E}$, $\mathbf{I}$ is the identity matrix for residual connections, and $\mathbf{W}^{(u)} \in \mathbb{R}^{d \times d}$ are the parameters of the $u$-th layer. A final residual connection is used to obtain the final node embeddings $\mathbf{F}'_v$:

$$\mathbf{F}'_v = \mathbf{F}_v + \mathbf{F}_v^{(U)}, \quad (6)$$

where $U$ denotes the number of layers in the GCN. These node embeddings, extracted from retrieved reports, are consistent with expert expressions and clearly distinguish normal from abnormal cases. Initialized by pre-trained DistilGPT2, they enhance the decoder's ability to capture nuanced details, leading to the generation of accurate, detailed radiology reports.

## 3.3 Graph-Enhanced Visual Representation

Recently, Segment Anything Model (SAM) has demonstrated powerful fine-grained Region of Interest capturing capabilities in general domains. Similarly, a series of works have emerged in the medical field, trained on medical data for medical image segmentation [33]. Considering that abnormal features in radiology images often manifest with fine granularity, we believe that the powerful fine-grained visual representation capability of the image encoder in MedSAM can be very beneficial for capturing subtle and detailed pathological changes in radiology images. Thus, we employ the image encoder (ViT) from MedSAM to encode radiology image $\mathcal{I}$, obtaining fine-grained patch embeddings $\mathbf{F}_I \in \mathbb{R}^{P \times d}$, where

$P$ represents the number of patch embeddings after linear projection. Notably, while IU-Xray generates radiology reports using two chest images (frontal and lateral), MIMIC-CXR uses a single image following R2Gen[2]'s annotation. For this, we use different linear projections for each dataset to project the extracted patch embeddings onto $P$ dimensions. Subsequently, the node embeddings $\mathbf{F}'_v \in \mathbb{R}^{N \times d}$ obtained in Subsection 3.2 are used to enhance the fine-grained patch embeddings $\mathbf{F}_I$. We employ Multi-Head Cross Attention (MHCA) to achieve this:

$$\hat{\mathbf{F}}_I = \text{MHCA}(\mathbf{F}_I, \mathbf{F}'_v),$$
$$\hat{\mathbf{F}}_v = \text{MHCA}(\mathbf{F}'_v, \mathbf{F}_I). \qquad (7)$$

Here, the patch embeddings $\mathbf{F}_I$ and node embeddings $\mathbf{F}'_v$ serve as queries, keys, and values for each other. The resulting enhanced patch and node embeddings are denoted as $\hat{\mathbf{F}}_I$ and $\hat{\mathbf{F}}_v$, respectively. Following this, the outputs of the MHCA are concatenated to leverage the strengths of both image and graph representations:

$$\hat{\mathbf{F}}_X = [\hat{\mathbf{F}}_v; \hat{\mathbf{F}}_I], \qquad (8)$$

where $\hat{\mathbf{F}}_X \in \mathbb{R}^{(N+P) \times d}$ is used to guide the text decoder in generating reports.

## 3.4 Graph-Guided Text Generation

In the image captioning task, the generative approach, often referred to as a captioner, excels in producing combined image-text representations that are crucial for vision-language understanding. This approach particularly shines in tasks requiring natural language generation. The captioner leverages a conventional encoder-decoder architecture: the image encoder, which could be a Vision Transformer (ViT) or ResNet, generates latent patch embeddings, while the text decoder autoregressively predicts tokenized texts based on the contextual information provided by the patch embeddings. The prediction process, introduced in Subsection 3.1, maximizes the conditional likelihood of the paired text using forward autoregressive factorization. Training this encoder-decoder with teacher-forcing allows for parallel computation, enhancing learning efficiency. In our work, the Graph-Enhanced Visual Representations $\hat{\mathbf{F}}_X$ obtained from Subsection 3.3 replace the patch embeddings in conventional encoder-decoder models. These enhanced visual representations are introduced to the decoder through a Multi-Head Cross-attention Module, which is randomly initialized and placed between the Masked Multi-Head Self-Attention and feed-forward neural network modules in each decoder layer, initialized by Distil-GPT2 [31]. The decoder then autoregressively generates the report from the Graph-Enhanced Visual Representations $\hat{\mathbf{F}}_X$. All details regarding the model architecture, graph construction, training, and parameters for text generation will be extensively discussed in Section 4.

## 4 EXPERIMENT

### 4.1 Dataset

We evaluate our proposed DCG on two widely-used radiology reporting benchmarks, IU-Xray [9] and MIMIC-CXR [10]. Following setting in [2], we divided and pre-processed both datasets for a balanced comparison.

**IU-Xray** [9] contains 7,470 chest X-ray images and 3,955 reports. Each report is associated with either one or both frontal and lateral view images. In line with [2], cases with a single image were omitted, leaving 2,069 training, 296 validation, and 590 testing cases.

**MIMIC-CXR** [10], currently the most extensive radiology dataset, includes 368,960 chest X-ray images and 222,758 reports and comes with an official split. Using the official data splits, which allocate 70%, 10%, and 20% for the training, validation, and test sets respectively, we have 258,272 cases in the training set, 36,896 in the validation set, and 73,792 in the test set.

During the construction of the organ-disease graph, our predefined organs align with those in [2]. However, in contrast to their approach of providing a rough estimate of about 20 disease findings related to organs/tissues, we rigorously differentiate between disease-specific and disease-free entities, resulting in a higher number of nodes. Across the entire dataset, IU-Xray features 191 nodes, while MIMIC-CXR comprises 276 nodes.

### 4.2 Implementation Details

**Image Encoder** Unlike previous works that use a ResNet-101 or DenseNet-121 pre-trained on ImageNet as the image encoder [2, 23, 28], we choose MedSAM's ViT [33] as our image encoder, excluding the MLP neck to extract patch embeddings. With an input image size of 512x512, MedSAM's ViT produces a feature map of 32x32x768, which is then flattened to a 1024x768 patch embedding. In alignment with established methods [1, 2], we process paired images for IU-Xray and a single image for MIMIC-CXR. To maintain consistency, we project the number of extracted patch embeddings from 1024 down to 256.

**Graph Construction** For graph generation, we utilize only the reports corresponding to the most similar images (based on cosine similarity) retrieved by BioMedCLIP [8] for constructing the graph. Following the predefined knowledge as detailed in [36], we preprocess and segment the reports, then extract predefined lists of organs and diseases using string matching through the Natural Language Toolkit (NLTK) [37]. Notably, we detect the presence of "no" and "normal" in sentences to differentiate between disease-free and disease-specific cases. Subsequently, pre-trained DistilGPT2 [31] initializes the Text decoder. After removing the LM Head, it is used to extract all disease-free and disease-specific entities as Node Embeddings, maintaining a dimension of 768. For different datasets, IU-Xray and MIMIC-CXR, the number of node embeddings is 191 and 276, respectively.

**Text Decoder and Generation** The pre-trained DistilGPT2 is also employed as the Text Decoder in our system. Our vocabulary encompasses DistilGPT2's tokens, supplemented with [BOS] and [EOS] tokens. In line with previous CXR report generators like Chen et al. [2], we standardized the format of ground-truth reports. This standardization includes limiting reports to 60 words, converting all text to lowercase, removing special characters, and substituting infrequent words with an unknown token. During testing, the decoder is capable of generating up to 128 subwords. For report generation, we apply beam search with a beam size of four, and during validation, a beam size of one is used for greedy search.

**Optimizing parameters** The model is trained on 4 NVIDIA 4090 GPUs, with a batch size of 16, for 20 epochs on both datasets.

**Table 1: The performances of our proposed DCG compared with other state-of-the-art systems on IU-Xray and MIMIC-CXR dataset. The baseline represents the simplest Encoder-Decoder structure we have implemented, with specific settings detailed in Section 4.**

| Dataset | Method | Avenue | NLG Metric | | | | | | | CE Metric | | |
|---|---|---|---|---|---|---|---|---|---|---|---|---|
| | | | BLEU-1 | BLEU-2 | BLEU-3 | BLEU-4 | ROUGE-L | METEOR | CIDEr | Precision | Recall | F1 |
| IU-Xray | R2Gen [2] | EMNLP'20 | 0.470 | 0.304 | 0.219 | 0.165 | 0.371 | 0.187 | - | - | - | - |
| | CMN [23] | ACL'21 | 0.475 | 0.309 | 0.222 | 0.170 | 0.375 | 0.191 | - | - | - | - |
| | PPKED [7] | CVPR'21 | 0.483 | 0.315 | 0.224 | 0.168 | 0.376 | 0.190 | 0.351 | - | - | - |
| | METrans [34] | CVPR'23 | 0.483 | 0.322 | 0.228 | 0.172 | 0.380 | 0.192 | 0.435 | - | - | - |
| | MMTN [35] | AAAI'23 | 0.486 | 0.321 | 0.232 | 0.175 | 0.375 | - | 0.361 | - | - | - |
| | DCL [1] | CVPR'23 | - | - | - | 0.163 | 0.383 | 0.193 | **0.586** | - | - | - |
| | **Baseline** | | 0.485 | 0.326 | 0.230 | 0.170 | 0.392 | 0.196 | 0.537 | - | - | - |
| | **Ours DCG** | | **0.514** | **0.330** | **0.241** | **0.186** | **0.401** | **0.211** | 0.578 | - | - | - |
| MIMIC-CXR | R2Gen [2] | EMNLP'20 | 0.353 | 0.218 | 0.145 | 0.103 | 0.277 | 0.142 | - | 0.333 | 0.273 | 0.276 |
| | CMN [23] | ACL'21 | 0.353 | 0.218 | 0.148 | 0.106 | 0.278 | 0.142 | - | 0.334 | 0.275 | 0.278 |
| | PPKED [7] | CVPR'21 | 0.360 | 0.224 | 0.149 | 0.106 | 0.284 | 0.149 | 0.237 | - | - | - |
| | METrans [34] | CVPR'23 | 0.386 | 0.250 | 0.169 | 0.124 | 0.291 | 0.152 | 0.362 | 0.364 | 0.309 | 0.311 |
| | MMTN [35] | AAAI'23 | 0.379 | 0.238 | 0.159 | 0.116 | 0.283 | - | 0.161 | - | - | - |
| | DCL [1] | CVPR'23 | - | - | - | 0.109 | 0.284 | 0.150 | 0.281 | **0.471** | 0.352 | 0.373 |
| | **Baseline** | | 0.368 | 0.235 | 0.156 | 0.106 | 0.289 | 0.145 | 0.391 | 0.371 | 0.316 | 0.319 |
| | **Ours DCG** | | **0.397** | **0.258** | **0.166** | **0.126** | **0.295** | **0.162** | **0.445** | 0.441 | **0.414** | **0.404** |

We select the checkpoint with the highest CIEDr metric for testing. The initial learning rates are set to $1e-5$ for the encoder and $1e-4$ for the other parameters, with all remaining AdamW hyper-parameters kept at their default values.

### 4.3 Evaluation Metrics

We evaluate performance using Natural Language Generation (NLG) metrics like CIDEr [38], BLEU [39], ROUGE-L [40], and METEOR [41], as well as Clinical Efficacy (CE) metrics. While BLEU measures n-gram overlap and is prone to textual bias, CIDEr more effectively assesses MRG systems by focusing on topic terms. For CE, we use the CheXPert tool [30] to label reports in 14 medical terminologies and assess clinical correctness with F1-Score, Precision, and Recall. However, CE metrics are applied only to the MIMIC-CXR dataset [10], as IU-Xray does not utilize CheXPert for labeling.

## 5 EXPERIMENT RESULTS

### 5.1 Comparison with State-of-the-art

To demonstrate the effectiveness of our proposed method, we compared its performance with various state-of-the-art models on two widely-used Radiology Report Generation (RRG) benchmarks: IU-Xray and MIMIC-CXR, as shown in Table 1. The models we compared include the classic R2Gen [2], a widely-used baseline for RRG; CMN [23] and PPKED [7], which integrate organ-disease graphs; and recent models such as METrans [34], MMTN [35], and DCL [1]. Our DCG outperforms other existing methods in both Natural Language Generation (NLG) and Clinical Effectiveness (CE) metrics. BLEU [39] measures the n-gram overlap between the predicted and actual reports. ROUGE-L [40] and METEOR [41], on the other hand, respectively represent the longest common subsequence between the generated and ground truth reports, and the harmonic mean of precision and recall, considering synonymy and paraphrasing.

Notably, a high CIDEr score indicates that the reports generated by our method have enhanced semantic richness and relevance to the clinical context.

### 5.2 Ablation Study

In this section, we first analyze the distribution and occurrences of normal and abnormal disease instances in the IU-Xray and MIMIC-CXR datasets, demonstrating the necessity of distinguishing between disease-specific and disease-free entities, as shown in Figure 4. Then, we assess the performance of off-line image retrieval using BioMedCLIP [8], as illustrated in Figure 5. Subsequently, we conduct ablation studies on IU-Xray to investigate the contribution of each component in our proposed DCG. Table 2 displays the impact of different visual encoders on accuracy, while Table 3 shows how various node encoders, node modeling methods, and information fusion approaches affect accuracy.

**Dataset Distributions**: We start by analyzing the distribution of the dataset from the disease entities pre-extracted from reports in the IU-Xray and MIMIC-CXR datasets, as shown in Figure 4. From the left part, it can be observed that the number of disease-free entities in IU-Xray is significantly higher than disease-specific entities, while the numbers of disease-free and disease-specific entities in MIMIC-CXR are unexpectedly balanced. We attribute this to our deliberate splitting of sentences such as "No pneumothorax, pleural effusion, or focal air space consolidation". From the right part, it can be observed that the most frequently occurring entities are "normal" and common diseases such as "pneumothorax" and "effusion", while disease-specific entities exhibit a long-tailed distribution, consistent with observations in the field.

**Retrieval Performance**: To verify that our proposed DCG-enhanced method indeed benefits from the constructed graph, we evaluated the degree of matching between the retrieved reports and

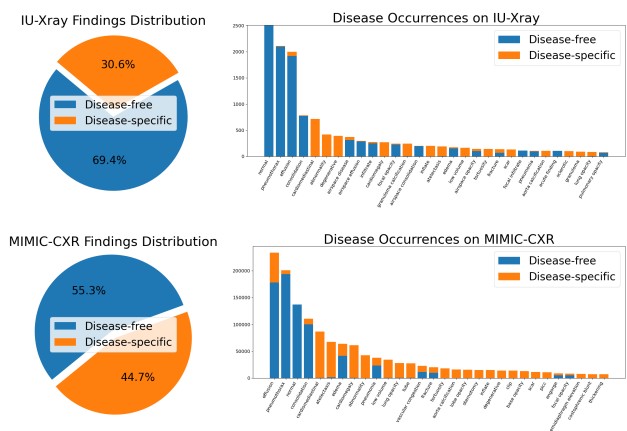

**Figure 4: Left part: Distribution of disease-specific and disease-free findings in IU-Xray and MIMIC-CXR reports; Right part: Frequency of occurrence of different types of diseases in IU-Xray and MIMIC-CXR reports.**

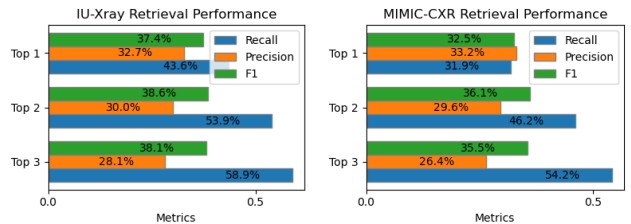

**Figure 5: Image Retrieval Performance using BioMedCLIP on the training sets of IU-Xray and MIMIC-CXR, respectively. Specific evaluation methods are detailed in Subsection 5.2.**

**Table 2: Ablation study of the visual encoder. (a) is the Vision Transformer (ViT) pretrained on BioMedCLIP [8]; (b) is the Convolutional Vision Transformer (CvT) pretrained on ImageNet-21K; (c) is the ViT fine-tuned on medical image segmentation using MedSAM [33].**

| Settings | Image Encoder | | Size | NLG Metric | | |
|---|---|---|---|---|---|---|
| | Model | Pretrained | | BLEU-4 | ROUGE-L | CIDEr |
| (a) | ViT-B/16 | BioMedCLIP[8] | 224 | 0.163 | 0.378 | 0.422 |
| (b) | CvT | ImageNet21k[42] | 384 | 0.165 | 0.379 | 0.426 |
| (c) | ViT-B/16 | MedSAM[33] | 512 | **0.170** | **0.392** | **0.537** |

the actual reports by performing image retrieval using BioMedCLIP [8] on input images. Since we pre-built disease-free and disease-specific pairs for each report, we could directly calculate the recall, precision, and F1 values for each retrieved report and actual report. The average results are shown in Figure 5. The results align with intuition: increasing the number of retrieved reports significantly improves disease recall while slightly decreasing precision. When the number of retrieved chest X-rays reaches 3, over 50% of entities are recalled on both datasets.

**Table 3: Ablation study of the node encoder and information fusion method, detailed in Subsection 5.2.**

| Settings | Node | Information Fusion | | NLG Metric | | |
|---|---|---|---|---|---|---|
| | Encoder | GCN | MHCA | BLEU-4 | ROUGE-L | CIDEr |
| Baseline | | | | 0.170 | 0.392 | 0.537 |
| (a) | PubMedBERT[43] | ✓ | ✓ | 0.165 | 0.376 | 0.426 |
| (b) | DistilGPT2 | | ✓ | 0.170 | 0.379 | 0.490 |
| (c) | DistilGPT2 | ✓ | | 0.172 | 0.395 | 0.509 |
| (d) Default | DistilGPT2 | ✓ | ✓ | **0.186** | **0.401** | **0.578** |

**Visual Encoder**: Good visual representations are crucial for the quality of radiology report generation. To this end, we investigated several state-of-the-art image encoders trained on both medical and general datasets, as shown in Table 2. It can be observed that there is not much difference in performance between (a) ViT-B/16@224 initialized with BioMedCLIP [8] and (b) CvT@384 pretrained on ImageNet21k (0.422 vs 0.426). (c) MedSAM [33] achieved significantly better results on the IU-Xray dataset compared to other image encoders, with a CIDEr score of 0.537. It is worth noting that BioMedCLIP was trained using a contrastive learning approach with 15 million medical image-text pairs, while MedSAM was trained on 100k images segmented. This suggests that fine-grained ROI priors may be more helpful in distinguishing different chest X-rays.

**Node Encoding and Information Fusion**: Since the constructed disease graph consists entirely of text, it is natural to use text encoders to encode these text nodes. In contrast to previous work [1] that used SciBERT to encode entities into node embeddings, we attempted (a) PubMedBERT, which aligns with BioMedCLIP [8] for multi-modal alignment; (b), (c), and (d) utilized DistilGPT2, with the decoder's initialization weights and vocabulary being consistent. Comparing the results, although (a) also introduced graph priors, it did not benefit report generation; in fact, the CIDEr score decreased from 0.537 to 0.426. We speculate that the IU-Xray dataset is too small to allow the model to establish associations between the node embeddings encoded by PubMedBERT and the token embeddings of DistilGPT2. (b) does not use graph convolutional networks to model organ-disease relationships. (c) does not utilize multi-head cross-attention to interact between all node embeddings and patch embeddings. (d) represents our final configuration. From the comparison results, it can be seen that each of our designs is crucial for radiology report generation.

## 5.3 Case Study

To further investigate the effectiveness of our proposed method, we conducted experiments on the IU-Xray [9] and MIMIC-CXR [10] datasets regarding the retrieved disease-free and disease-specific pairs, our baseline, and our proposed DCG. Descriptions containing disease-free and disease-specific pairs are highlighted in green and blue, respectively. The gray text indicates descriptions that did not match in the retrieval report. It can be observed that the IU-Xray dataset contains some omitted information such as "XXXX", which may affect the generation performance. However, to maintain consistency with previous methods, we did not clean these sentences. The orange text indicates that despite augmenting the predefined disease types, the MIMIC-CXR dataset still contains some disease descriptions that are low in frequency and not defined

| Image | Ground Truth Report | Retrieved Pairs | Baseline | Ours |
|---|---|---|---|---|
| **IU-Xray** | The heart is normal in size. There is a round density in the AP XXXX. XXXX study performed in XXXX is not available for review at this time. Lungs are hyperinflated with flattened diaphragms. Calcified right lower lobe granuloma. No focal airspace consolidation, pneumothorax, or pleural effusion. No pulmonary edema. No acute bony abnormality. | **Disease-free**: normal-airspace no pneumothorax-pleural no airspace consolidation-airspace no edema-lung no airspace effusion-airspace no abnormality-bone **Disease-specific**: inflate-lung diaphragm flatten-mediastinum granuloma calcification-lung | The heart size and pulmonary vascularity appear within normal limits. The lungs are free of focal airspace disease. No pleural effusion or pneumothorax is seen. Degenerative changes are present in the spine. | The heart size is normal. The lungs are normally inflated without evidence of focal airspace disease. No pleural effusion or pneumothorax is seen. There is a calcified granuloma in the right upper lobe. No acute bony abnormality. |
| **MIMIC-CXR** | Ap upright and lateral views of the chest were obtained. Elevated right hemidiaphragm is again noted. Mild cardiomegaly is also stable. There is no focal consolidation effusion or overt signs of CHF. Mediastinal contour is stable. Bony structures are intact. A mild scoliosis is again noted with a superior end plate compression deformity. | **Disease-free**: obtain-views lateral no effusion-pleural no consolidation-lung no pneumothorax-pleural stable-mediastinum **Disease-specific**: hemidiaphragm elevate-mediastinum cardiomegaly-heart intact-bone scoliosis-spine deformity-spine | The lungs are clear without focal consolidation. No pleural effusion or pneumothorax is seen. The cardiac and mediastinal silhouettes are stable. | Ap upright and lateral views of the chest provided. There is no focal consolidation effusion or pneumothorax. The cardiomediastinal silhouette is normal. Imaged osseous structures are intact. No free air below the right hemidiaphragm is seen. |

| **Color meanings:** | Green: Disease-Free Description. | Gray: Non-Related Description. | Green with underline: Correctly generated Disease-Free Description. |
|---|---|---|---|
| | Blue: Disease-Specific Description. | Orange: Undefined Disease Description. | Blue with underline: Correctly generated Disease-Specific Description. Red with strikethrough: Erroneously Retrieved/Generated Description. |

**Figure 6: Illustrations of reports from ground truth, ours, and baseline (the simplest encoder-decoder) and retrieved disease-free and disease-specific pairs for two samples from IU-Xray [9] and MIMIC-CXR [10], respectively. For better visualization, different colors highlight different description types.**

by us, such as "CHF", which stands for Congestive Heart Failure. The quantities of different types of pairs from IU-Xray and MIMIC-CXR confirm that the distribution statistics are generally consistent, with slightly more normal descriptions than abnormal ones in IU-Xray, while the numbers of normal/abnormal descriptions are balanced in MIMIC-CXR. The baseline is the simplest encoder-decoder structure, and it tends to predict higher-frequency occurrences influenced by data bias, such as "No pleural effusion" or "Degenerative changes", regardless of whether these diseases appear in the images. In contrast, our proposed DCG generates corresponding descriptions for accurately retrieved organ-disease pairs, even including some relatively low-frequency cases such as "Calcified granuloma" or "Lung inflated". However, it is worth mentioning that in the examples from MIMIC-CXR, our proposed DCG retrieved "No pneumothorax-pleural", which led to the generation of "No pneumothorax" in the report, not mentioned in the ground truth report. According to our observations, "No pneumothorax", "No effusion", and "No consolidation" often appear together, indicating that our generated report is likely correct, but due to the omission of the normal description "No pneumothorax" by the physician, our generated report was not properly evaluated, consistent with the situation described in Section 1. These circumstances point us toward the development direction of radiology report generation tasks: 1) Dataset cleaning and establishing uniform baseline standards are crucial for evaluating all RRG methods. 2) Further expanding the knowledge graph is

essential for improving the performance of RRG methods. 3) Exploring evaluation methods beyond n-grams is necessary for advancing RRG research.

## 6 CONCLUSION AND DISCUSSION

In this paper, we present a novel approach for constructing organ-disease graphs in radiology report generation tasks. Addressing the issue where existing methods cover only a limited range of disease types and do not fully align with actual physician-written reports, especially in identifying normal and abnormal attributes of diseases, we introduced a Divide-and-Conquer method, called DCG. This method relies on similarity-based report retrieval to build fine-grained organ-disease graphs for each report, strictly categorizing nodes as *disease-free* or *disease-specific*, based on their normal or abnormal attributes and specific locations. Such a strategy helps to differentiate between normal and abnormal conditions, reducing the impact of bias due to the prevalence of common diseases. Experiments on two popular benchmarks verify the effectiveness of our method in generating accurate and meaningful reports.

Despite these promising results, we acknowledge that our method has room for improvement: (1) report retrieval is conducted offline within each dataset, and (2) the retriever has not been fine-tuned specifically for chest X-rays. We aim to address these issues in future work, and we hope that our constructed graph will further enhance the performance of subsequent radiology report generation tasks.

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
