# OpenReview forum: "Divide and Conquer: Isolating Normal-Abnormal Attributes in Knowledge Graph-Enhanced Radiology Report Generation"
_acmmm.org/ACMMM/2024/Conference — MM2024 Poster_

### Official Review · Reviewer_jxhA · 2024-05-02

**Rating:** 4
**Confidence:** 2

**Summary:**

The article proposes a new Divide-and-Conquer method for constructing more comprehensive organ disease maps and distinguishing between disease-specific and disease-free nodes, with innovative and practical applications for medical report generation tasks.

**Strengths:**

1. Innovation and application value: The article proposes a new Divide-and-Conquer method to construct a more comprehensive organ disease map, which is innovative and has practical application value.

2. Sufficient experiments: The article conducts sufficient experimental verification on two public datasets, demonstrating the effectiveness of the method, and using multiple metrics for evaluation, making the results more convincing.

3. Clear structure: The overall structure of the article is clear, with each section arranged logically and in line with academic paper writing norms.

**Limitations:**

1. How many previously uncovered entities can be captured by the entity extraction method used in the paper compared to previous methods? The proposed method does not seem to address well the challenge of "Limited disease coverage" mentioned in the introduction. It is more about distinguishing between normal and abnormal disease conditions.

2. The proposed method involves multiple modules and networks, although the performance is improved, how does it compare to baseline in terms of training time and inference speed?

**Suitability:**

2

---

### Official Review · Reviewer_1s5s · 2024-05-23

**Rating:** 4
**Confidence:** 3

**Summary:**

Compared to image captioning tasks, the medical report generation task presents additional challenges due to the more subtle differences between images and the specialized, complex medical terminology used in reports. This paper addresses these challenges by constructing a comprehensive knowledge graph and employing a Divide-and-Conquer approach (DCG) to distinguish between normal and abnormal anatomical attributes. This enables the model to consciously focus on abnormal information and mitigate the impact of overly common diseases on report generation.

**Strengths:**

1. This paper identifies the challenges of automatically generating radiology reports and emphasizes that existing knowledge graphs cover only a limited and specific set of terms, an issue that has not been previously addressed.
2. The proposed Divide-and-Conquer approach (DCG) and the knowledge graph constructed from Disease-free pairs and Disease-specific pairs are innovative and reasonable. They help the model focus on abnormal information and mitigate the impact of long-tail distribution of disease labels.
3. Comprehensive experiments, including comparisons with state-of-the-art methods and ablation studies, provide evidence for the effectiveness of the proposed approach.

**Limitations:**

1. CvT-212 DistilGPT2 [1] uses the same backbone as this paper, but it does not employ a knowledge graph, and the performance difference is minimal. Could you provide insights into this observation?
2. Could more ablation studies on the proposed knowledge graph be provided to demonstrate its effectiveness and superiority?



[1] Nicolson A, Dowling J, Koopman B. Improving chest X-ray report generation by leveraging warm starting[J]. Artificial intelligence in medicine, 2023, 144: 102633.

**Suitability:**

3

---

### Official Review · Reviewer_QoDD · 2024-05-24

**Rating:** 3
**Confidence:** 2

**Summary:**

The paper presents a novel method for radiology report generation, addressing the limitations of current approaches that fail to differentiate between normal and abnormal attributes in radiology images. The proposed Divide-and-Conquer Graph (DCG) enhances the radiology report generation process by constructing comprehensive organ-disease graphs that distinguish between disease-free and disease-specific nodes. This approach improves the model’s ability to generate accurate clinical descriptions by focusing on both normal and abnormal attributes, thus enhancing the overall quality and applicability of the generated reports. Extensive experiments on benchmark datasets IU-Xray and MIMIC-CXR demonstrate the effectiveness of the proposed method.

**Strengths:**

The paper introduces a significant advancement in radiology report generation by addressing the critical issue of distinguishing between normal and abnormal attributes in medical images. The use of a Divide-and-Conquer strategy to construct a more comprehensive organ-disease graph is innovative and enhances the model's ability to generate accurate and clinically relevant reports. The proposed method leverages the powerful BioMedCLIP for retrieving similar radiology images, which helps in constructing fine-grained disease-specific and disease-free nodes. This approach is technically sound and theoretically justified, with rigorous experiments conducted on benchmark datasets demonstrating the superiority of the proposed method over existing state-of-the-art models. The clarity of the presentation, along with detailed explanations of the methodology and thorough evaluation, makes the paper a valuable contribution to the field of medical image analysis and natural language generation.

**Limitations:**

Despite the promising results, the paper has some limitations that need to be addressed. First, the offline retrieval process using BioMedCLIP for constructing the graph might not be scalable for large datasets, which could impact the efficiency of the proposed method in real-world applications. Second, the evaluation metrics primarily focus on n-gram overlap measures such as BLEU and ROUGE, which may not fully capture the clinical relevance and accuracy of the generated reports. Incorporating more clinically-oriented evaluation metrics could provide a better assessment of the model's performance. Additionally, the method's reliance on predefined organ-disease pairs may limit its ability to generalize to new or rare diseases not covered in the training data. Future work could explore more adaptive and scalable approaches for graph construction and integrate additional evaluation metrics to better capture the clinical utility of the generated reports. Furthermore, fine-tuning the retriever specifically for chest X-rays and exploring more advanced techniques for handling data bias could further improve the model's performance. Overall, while the proposed method shows significant improvements, addressing these limitations could enhance its applicability and robustness in clinical settings.

**Suitability:**

2

---

### Official Review · Reviewer_hFKi · 2024-05-25

**Rating:** 6
**Confidence:** 4

**Summary:**

The paper proposes a novel method (DCG) for generating radiology reports from medical images. It addresses the limitations of existing models by constructing a comprehensive knowledge graph that distinguishes between normal and abnormal attributes, enhancing the accuracy of the generated reports. The approach involves retrieving similar radiology images and their reports to build disease-free and disease-specific nodes, which improve the correlation between visual representations and medical terminology. Extensive experiments on the IU-Xray and MIMIC-CXR datasets demonstrate that DCG outperforms previous methods in generating clinically accurate and detailed reports.

**Strengths:**

The paper introduces a more detailed and comprehensive approach to creating knowledge graphs that distinguish between normal and abnormal conditions, which enhances the model’s ability to generate precise and meaningful radiology reports. The proposed strategy of separating disease-free and disease-specific nodes helps the model to focus on critical abnormalities, improving the accuracy of the generated reports. I highly recommend the proposed method to extend existing knowledge with more specific attributes. Then, by improving the alignment between visual data and medical terminology, the proposed method ensures more accurate and contextually relevant report generation. The whole paper is very well written and organized. The code is provided with the submission which makes it easy to reproduce the results. I have checked the code that is clean and readable. The paper conducts thorough experiments on well-known datasets (IU-Xray and MIMIC-CXR), showing that their method outperforms current leading techniques in generating accurate and clinically relevant reports.

**Limitations:**

I have one concern, since the BiomedCLIP is pre-trained on the whole MIMIC-CXR datasets, will it be fair to be used to retrieve similar reports?

**Suitability:**

3

---

### Meta-Review · Area_Chair_9Rmv · 2024-07-02

**Recommendation:** Accept (Poster)
**Confidence:** 4

**Metareview:**

Overall, all reviewers are satisfied with the response given by the authors, and are glad to see that the quality of the paper has been improved substantially.